# Metabolic Reprogramming at the Edge of Redox: Connections Between Metabolic Reprogramming and Cancer Redox State

**DOI:** 10.3390/ijms26020498

**Published:** 2025-01-09

**Authors:** José J. Serrano, Miguel Ángel Medina

**Affiliations:** 1Departamento de Biología Molecular y Bioquímica, Facultad de Ciencias, Universidad de Málaga, E-29071 Málaga, Spain; josejoaquinserranomorales1984@gmail.com; 2Instituto de Investigación Biomédica y Plataforma en Nanomedicina IBIMA Plataforma BIONAND (Biomedical Research Institute of Málaga), E-29071 Málaga, Spain; 3CIBER de Enfermedades Raras (CIBERER, Spanish Network of Research Center in Rare Diseases), Instituto de Salud Carlos III, E-28029 Madrid, Spain

**Keywords:** redox, cancer, metabolism, feedback, emergence, reprogramming

## Abstract

The importance of redox systems as fundamental elements in biology is now widely recognized across diverse fields, from ecology to cellular biology. Their connection to metabolism is particularly significant, as it plays a critical role in energy regulation and distribution within organisms. Over recent decades, metabolism has emerged as a relevant focus in studies of biological regulation, especially following its recognition as a hallmark of cancer. This shift has broadened cancer research beyond strictly genetic perspectives. The interaction between metabolism and redox systems in carcinogenesis involves the regulation of essential metabolic pathways, such as glycolysis and the Krebs cycle, as well as the involvement of redox-active components like specific amino acids and cofactors. The feedback mechanisms linking redox systems and metabolism in cancer highlight the development of redox patterns that enhance the flexibility and adaptability of tumor processes, influencing larger-scale biological phenomena such as circadian rhythms and epigenetics.

## 1. Introduction

Metabolic reprogramming has recently been established as a hallmark of cancer [1,2], though the role of metabolism in tumor development has been recognized for over a century [3,4,5,6]. Its significance, however, was overshadowed by the discovery of DNA’s structure and the subsequent focus on identifying genes responsible for cancer and other diseases. A substantial portion of cellular metabolism relies on active redox elements, which are essential for the various pathways comprising the metabolic network. The reprogramming process is not always driven by genetic factors, as seen with oncometabolites [7,8]. In proliferative tumor cells, the availability of alternative electron acceptors and oxygen consumption may play pivotal roles in metabolic reprogramming [9]. While the metabolic activity of tumors, particularly regarding glucose, fatty acids and glutamine fluxes, is well-documented [10], the precise roles of redox systems in tumor metabolic reprogramming remain to be fully elucidated and redox-sensitive mechanisms could be better characterized at a molecular level. This review explores the most significant connections described to date between metabolic reprogramming and the redox state in cancer.

## 2. Glucose Metabolism and Cancer Redox State

The glycolytic pathway is the most extensively studied process associated with tumor progression and metabolic reprogramming in cancer (Figure 1). It traces back to Otto Warburg’s groundbreaking work in the early 20th century, in which he observed that tumors, even under normoxic conditions, rely on fermentative glycolysis for energy production [3,4]. Warburg later hypothesized that this shift was due to a defective electron transport chain (ETC) in tumor cells, leading to the excessive use of glycolysis [11]. However, this hypothesis was rejected when evidence showed that tumor tissues supplemented with NAD+ maintained functional ETC activity [12].

Rather than inhibiting ETC, tumor cells appear to regulate its activity, enhancing antioxidant capacity and diversifying the metabolic substrates available as fuels. When oxidative phosphorylation predominates, its contribution is controlled by metabolic conditions driven by the tumor microenvironment [6,13]. Nevertheless, mitochondrial metabolism in tumor cells is generally altered, disrupting intracellular and local reactive oxygen species (ROS) levels. To counteract these disruptions, tumor cells often increase glucose metabolism through the pentose phosphate pathway (PPP), which boosts the production of reducing equivalents like NADPH (Figure 2). NADPH, in turn, supports the synthesis of glutathione (GSH) and thioredoxin (Trx), essential for maintaining redox homeostasis. For instance, a study demonstrated that inhibiting glucose-6-phosphate dehydrogenase (G6PD) in the PPP, combined with Trx system suppression, significantly elevates oxidative stress in tumor cells [14].

Enzymes in the non-oxidative branch of the PPP, such as transketolase (TLK), play critical roles in redox balance. TLK knockout has been shown to decrease NADPH levels while increasing glycolytic intermediates like fructose-1,6-bisphosphate. This, in turn, activates tumor cell pyruvate kinase M2, which facilitates oxidative phosphorylation by restoring Krebs cycle activity. The expression of TLK and other PPP enzymes is regulated by cellular ROS levels through Nrf2 activation, which transcriptionally modulates genes like *TLK* and *G6PD*, helping cells counter oxidative stress [15]. Similar redox control mechanisms involving the PPP have been observed in yeast, independent of NADPH production, which may enhance our understanding of tumor metabolic versatility and heterogeneity [16].

In cancer cells, the existence of two high-rate PPP pathways has been demonstrated [17]. During the G1-to-S phase transition, reduced ETC flow appears to modulate ATP production, shifting tumor reliance from oxidative phosphorylation to glycolysis, which may help prevent ROS overproduction from the ETC. This interplay between redox regulation and the Warburg effect has also been linked to autophagy in tumorigenesis [18,19].

The dynamic switch between oxidative phosphorylation and glycolysis is also involved in drug resistance and phenotypic changes in cancer. For example, in docetaxel-resistant prostate cancer cells, this metabolic transition enhances oxidative phosphorylation and promotes a proinvasive phenotype with increased proliferation and migration. Reversing this shift by reexpressing miR-205—a microRNA suppressed in resistant cells—restores sensitivity to docetaxel and rebalances metabolism toward glycolysis [20]. Similarly, some tumor cells adapt to genotoxic stress by modulating the OXPHOS and glycolytic pathways. For instance, mTOR translocates to mitochondria, interacting with hexokinase II to suppress glycolytic flow. This reduces hexokinase II’s interaction with MnSOD, freeing MnSOD to perform its ROS-scavenging function [21].

Heterogeneity in oxidative stress management has been observed across breast cancer cell lines, underscoring the diversity in redox adaptations across cancer types [22]. Furthermore, metabolic reprogramming that increases glycolytic dependence has been shown to sustain redox homeostasis during metastasis in breast cancer cells [23]. On the other hand, it should always be kept in mind that the metabolic regulation of cancer cell lines can be very different from the same tissue type in vivo.

## 3. Glutamine Metabolism and Cancer Redox State

Glutamine plays a pivotal role in sustaining oxidative phosphorylation (OXPHOS) in tumor cells, acting as a crucial metabolite in cancer progression. By generating α-ketoglutarate, glutamine replenishes the Krebs cycle, ensuring the continuous production of NADH and FADH_2_. These molecules serve as electron donors for the ETC, creating the proton electrochemical gradient required for ATP synthesis via OXPHOS. Under hypoxic conditions, α-ketoglutarate can also be converted into citrate, which generates NADPH through the malic enzyme I reaction, supporting redox homeostasis [24]. This reductive tricarboxylic acid (TCA) cycle (Figure 3) has emerged as a metabolic vulnerability, offering potential for targeted therapies in conditions like acute myeloid leukemia [25].

The significance of glutamine in OXPHOS was highlighted in studies examining the ATP production driven by Akt and Ras oncogenes. These studies revealed that ATP generation was not significantly increased by glycolytic activation alone but was largely dependent on α-ketoglutarate derived from glutamine. Additionally, glutamine contributes to redox balance by elevating the NADH/NAD^+^ ratio, even under hypoxic conditions [26]. A possible link between the Akt activity level and mTOR translocation to the mitochondria, influencing glutamine metabolism and enhancing OXPHOS while preserving redox homeostasis, remains an area for further exploration. This underscores how tumor cells adapt their redox state to respond to external and internal challenges.

Glutamine’s role in redox regulation is further evidenced by studies on leukemia cell lines, where glutamine deprivation significantly reduced glutathione (GSH) levels and mildly elevated reactive oxygen species (ROS) levels [27]. In pancreatic ductal adenocarcinoma (PDAC) cells, combining glutaminase inhibitors with H_2_O_2_ heightened their sensitivity to ROS. This effect required not only glutamine-derived compounds but also non-essential amino acids, relying on the mitochondrial (GOT2) and cytosolic (GOT1) aspartate aminotransferases to generate NADPH and maintain the redox balance, particularly in tumors expressing KRAS [28].

The importance of glutamine becomes even clearer in tumor cells harboring mutations in Krebs cycle enzymes or defects in ETC complexes I and III. In such cases, glutamine metabolism shifts toward reductive carboxylation mediated by isocitrate dehydrogenase I, a process that demands elevated mitochondrial NADPH levels. This is achieved through enzymes like nicotinamide nucleotide transhydrogenase, which transfers electrons from NADH to NADPH [29,30]. The reductive carboxylation of glutamine also supports lipogenesis in the cytosol of hypoxic tumor cells [31] and helps tumor cells adapt to anchorage independence by mitigating ROS elevation through NADPH production [32].

However, the dependence of cancer cells on glutamine for redox regulation varies across tumor types. Tumor cells “addicted” to glutamine exhibit increased ROS levels and higher glucose uptake when glutamine is absent, though glucose is not utilized for anaplerosis. In contrast, non-addicted tumor cells experience a drop in GSH levels upon pyruvate transporter inhibition, potentially inducing a metabolic shift toward glutamine addiction to manage oxidative stress. This is likely achieved by channeling glucose into pathways like the pentose phosphate pathway or serine metabolism, which generate reducing equivalents [33,34].

Despite the weight of evidence linking glutamine metabolism to redox control, some studies suggest the opposite: certain tumor cells shift their metabolic phenotype toward OXPHOS at the expense of glycolysis [35]. Therefore, further investigation into the context-specific roles of glutamine in cancer metabolism is required.

## 4. Serine Metabolism and Cancer Redox State

Several other metabolic pathways contribute significantly to the generation of reducing equivalents, including those directly linked to glucose or glutamine metabolism, such as serine metabolism (Figure 4) [36,37,38]. The role of serine in carcinogenesis has gained particular attention due to its involvement in nucleotide synthesis through the production of one-carbon units (e.g., formate) in the mitochondria, which are essential for cell proliferation [39]. These one-carbon units participate in DNA and RNA methylation processes by regenerating methionine from homocysteine [40].

Studies on colon cancer cells have revealed that serine metabolism couples *de novo* nucleotide synthesis with glutathione production, highlighting its connection to redox homeostasis [41]. This is further supported by the fact that one-carbon metabolism is a major source of NADPH, a key molecule in maintaining redox balance [42]. Formate production in tumor mitochondria is inhibited when ETC complex I is suppressed, while normal fibroblasts remain unaffected by this inhibition [43]. Formate synthesis in mitochondria prioritizes ATP production through the MTHFD2 reaction. Formate can also be converted into 10-formyl-THF and subsequently into THF by the enzyme 10-formyl-THF dehydrogenase (ALDH1L2), which is often overexpressed in various cancers. This process generates NADPH, acting as a redox buffer to counter increased ROS levels from enhanced oxidative phosphorylation driven by NADH produced in the MTHFD2 reaction [37].

In MDA-MB-231 breast cancer cells, the enzyme 3-phosphoglycerate dehydrogenase (PHGDH), which catalyzes the first step in serine synthesis from the glycolytic intermediate 3-phosphoglycerate, is overexpressed under hypoxic conditions. This overexpression may enable these cells to produce serine de novo, which, through the folate cycle, generates NADPH to maintain redox homeostasis. This mechanism appears to be particularly critical in tumor cells sensitive to ROS elevation, such as cancer stem cells [44]. Inhibiting PHGDH has been shown to suppress the growth of several breast cancer cell lines, primarily by disrupting redox balance. However, this effect is observed only in lines with PHGDH amplification, which represents a small subset of breast cancer cases [45]. Our group has proposed that the activating effect of dimethyl fumarate on glycolysis in endothelial cells, necessary for tumor angiogenesis, along with its inhibitory effect on OXPHOS, could result from the downregulation of serine and glycine synthesis due to PHGDH inhibition in these cells [46].

Beyond enzymatic control, serine metabolism is also regulated at the gene expression level. For instance, the recruitment of MDM2, a negative regulator of p53, to chromatin occurs when serine and glycine levels are limited [47]. Additionally, recent findings indicate that CDK12 overexpression drives tumorigenesis in breast cancer while simultaneously increasing susceptibility to therapies targeting one-carbon metabolism [48].

## 5. Parametabolic Regulation and Proline Metabolism

The significance of the redox state in regulating tumor metabolism can be understood within the framework of parametabolic regulation [49]. This concept suggests that the true importance of metabolic pathways lies not in their primary products but in the cofactors involved, which would be the essential drivers of cellular processes [50]. In this context, proline metabolism plays a pivotal role (Figure 5). The interconversion between proline and its oxidized form, pyrroline-5-carboxylate, facilitates the transfer of redox equivalents between the mitochondria and cytosol through the enzyme pair proline oxidase (POX)/proline dehydrogenase (PRODH). These enzymes, located on the inner mitochondrial membrane, can donate electrons to the ETC, contributing to the formation of reactive oxygen species (ROS) for intracellular signaling [51].

Proline exhibits a dual role in tumor growth. Its ability to enhance ROS production can promote apoptosis via intrinsic and extrinsic pathways. However, under specific conditions of metabolic stress or inflammation, POX can shift to favor survival [52]. This survival mechanism might involve the degradation of extracellular matrix collagen, a proline reservoir supporting the tumor microenvironment. Proline derived from collagen can help sustain the balance of cofactors like NAD and NADP, especially under high glucose consumption, paralleling the role of lactate in similar scenarios [53]. For instance, studies on PC9 lung cancer cells show that the proline–pyrroline-5-carboxylate cycle supports glycolytic metabolism. Inhibiting enzymes in this cycle reduces the levels of NAD^+^, NADH, NADP^+^ and NADPH without altering their ratios, emphasizing their parametabolic regulation [54]. Furthermore, this cycle links directly to glutamine metabolism, as proline can be synthesized from glutamine in an MYC-dependent manner, highlighting the oncogene’s role in maintaining the redox state through proline interconversion [55].

The enzymes driving the proline cycle, particularly POX/PRODH, are critical for ROS generation in the ETC. Notably, the proline cycle is connected to the succinate cycle, which regulates electron flow through the ETC and ROS production. POX/PRODH can transfer electrons directly to ubiquinone, sustaining short-term respiratory capacity. Meanwhile, succinate acts as an uncompetitive inhibitor of POX/PRODH, reducing ROS production, restoring respiratory capacity and stabilizing protein levels in ETC complexes, which are otherwise depleted by prolonged POX/PRODH activity [56]. This allows us to connect to the next key point in the control of cell redox state, the Krebs cycle.

## 6. The Krebs Cycle and Cancer Redox State

Three of the eight enzymes involved in the Krebs cycle (Figure 6) significantly influence the redox state of tumor cells and play a critical role in tumor progression [57]. Isocitrate dehydrogenase (IDH) is one such enzyme; when mutated, it disrupts redox homeostasis by lowering glutathione (GSH) levels, likely due to a reduction in available glutamate. Cells with mutations in *IDH1* and *IDH2* also exhibit increased NADPH oxidation, as the oxidative decarboxylation of isocitrate is impaired [58,59]. These mutations confer a novel catalytic activity to IDH, resulting in the production of (*R*)-2-hydroxyglutaric acid, an oncometabolite implicated in tumorigenesis [60,61]. Since 2016, multiple inhibitors targeting mutant *IDH1/2* have been approved as anticancer therapies, including vorasidenib (Voranigo, Servier Pharmaceuticals LLC., Suresnes, France), which received FDA approval on 6 August 2024 for treating grade 2 astrocytoma or oligodendroglioma with susceptible IDH mutations. For further details, refer to the FDA announcement: https://www.fda.gov/drugs/resources-information-approved-drugs/fda-approves-vorasidenib-grade-2-astrocytoma-or-oligodendroglioma-susceptible-idh1-or-idh2-mutation (accessed on 12 November 2024).

Succinate, another intermediate in the Krebs cycle and its associated enzyme, succinate dehydrogenase (SDH), have also been involved in tumor progression and redox regulation [62]. SDH is known to play a pivotal role in metabolic reprogramming in various cell types, including macrophages during inflammation. In such contexts, SDH drives a metabolic shift toward glycolysis for ATP production and increases succinate-dependent ROS production, promoting the expression of pro-inflammatory genes. This process is accompanied by heightened mitochondrial membrane potential, which generates redox signals that influence the activity of hypoxia-inducible factor-1α (HIF-1α) [63]. Additionally, the antitumor drug lonidamine has been shown to inhibit ubiquinone reduction by SDH, although its exact mechanism of action remains unclear [64].

Unlike the oxidative state observed with IDH or SDH deficiencies, fumarase (FH) deficiency leads to a more reduced cellular redox state due to the loss of its cytoplasmic isoform. This results in increased cytosolic fumarate levels, derived from purine nucleotide metabolism [65]. However, FH deficiency also induces oxidative stress through the formation of fumarate-glutathione (succinated-GSH) adducts. This drives metabolic flux toward GSH synthesis, increasing cystine uptake and depleting NADPH reserves as cystine is reduced to cysteine—an essential precursor for GSH formation [66].

## 7. Heme Catabolism and Cancer Redox State

Less explored branches of nitrogen metabolism also play a significant role in maintaining cellular redox balance and are increasingly recognized as critical to tumor progression. Among these, the degradation of the heme group and its conversion into bilirubin stands out. This process is driven by the sequential action of heme oxygenase-1 (HO-1) and biliverdin reductase (BVR), which catalyze the release of Fe^2+^ from heme and its subsequent export from the cell. This mechanism prevents the formation of reactive oxygen species (ROS) through the Fenton reaction, thereby offering protection against oxidative stress [67]. Additionally, the process results in the production of bilirubin, a potent antioxidant that scavenges hydroxyl radicals and protects against lipid peroxidation [68]. This antioxidative action complements that of the upstream reactions.

The interplay between bilirubin production and glutathione (GSH) metabolism further highlights their combined importance. GSH serves as the primary antioxidant in the aqueous cytosolic fraction, while bilirubin provides key protection in the lipid fraction of the plasma membrane [69]. This complementary antioxidant defense system may represent a critical mechanism in tumor cells for regulating oxidative stress and promoting tumor progression. It also underscores the redirection of metabolic pathways to prioritize the synthesis of intermediates vital for maintaining the balance of reducing equivalents. Both GSH and bilirubin production consume high amounts of NAD(P)H, underscoring the metabolic cost associated with sustaining these antioxidant systems.

## 8. Master Regulators of Cancer Redox State

The transcription factor NRF2 (Figure 7) was first isolated in 1994 [70] and subsequently identified in 1999 as a key cytoprotective factor against xenobiotics [71]. Since then, it has emerged as a central regulator of oxidative stress and toxicity [72]. Today, NRF2 is widely regarded as a master regulator of cellular redox homeostasis, orchestrating metabolic shifts to maintain redox balance [73,74,75]. Its protein levels are tightly regulated by KEAP1, a ubiquitin ligase scaffold protein that binds and ubiquitinates NRF2, targeting it for proteasomal degradation [75]. Under oxidative stress conditions, NRF2 escapes KEAP1-mediated degradation, accumulates in the nucleus and activates the transcription of numerous genes involved in antioxidant defense [76].

Many cancers exploit the KEAP1/NRF2 pathway through genetic alterations. Loss-of-function mutations in *KEAP1* or gain-of-function mutations in *NRF2* have been shown to promote tumorigenesis [75,77,78,79,80,81,82]. However, NRF2 also exhibits tumor-suppressive functions under certain contexts [83,84,85], highlighting its dual role in cancer depending on the cellular environment [86]. This complexity has been explored in studies dissecting NRF2′s specific contributions to the hallmarks of cancer [87]. As a result, the KEAP1/NRF2 pathway has garnered significant interest as a therapeutic target, with both activators and inhibitors of this axis showing potential in cancer treatment strategies [75,81,82]. Comprehensive reviews of the therapeutic implications of modulating this pathway have been recently published [87,88,89].

While NRF2 is the most studied member of the NRF family, other NRF proteins also play important roles. NRF1, for example, is a critical regulator of proteostasis, orchestrating the “proteasome bounce-back” response to maintain protein homeostasis in various diseases [90,91,92]. Its expression levels vary in different cancer types—being upregulated in some and downregulated in others—suggesting a dual role in cancer progression [93]. The shared and distinct roles of NRF1 and NRF2 have been reviewed extensively [94,95].

NRF3, another member of the family, was first cloned and characterized by Yamamoto’s group, which also studied NRF2 [96]. The differential and overlapping functions of NRF3, NRF2 and NRF1 have been investigated and revealed [94]. NRF3 promotes tumor growth by facilitating the degradation of tumor suppressors p53 and pRb through a ubiquitin-independent proteolysis mechanism mediated by the 20S proteasome [97,98]. Recent research reveals that NRF3 is inducible under arginine depletion and supports tumor growth via activation of the arginine-dependent mTORC1 signaling pathway [99].

## 9. Conclusions

The study of cancer metabolism has become a hotspot of cancer research in recent decades, driven by the revival of Warburg’s hypothesis and a growing recognition that genes play a less decisive role in the origin and evolution of cancer than previously thought. Increasing evidence, particularly from the field of epigenetics, reveals a deep connection between metabolism and genetics, mediated by the chemical interactions between redox pairs of molecules [100,101,102,103,104,105]. This insight highlights redox systems as underappreciated yet critical players in biological regulation. As discussed in the main text, numerous branches of cellular metabolism in cancer are regulated by dynamic shifts in redox pairs. However, cancer is often described as a multi-scale process, involving disruptions across various layers of biological organization, from molecular pathways to organismal physiology [106]. The prevalence of diverse redox patterns—spatiotemporal arrangements of redox molecules across biological scales—emphasizes their significant role in the origin and progression of cancer. Table 1 summarizes the main metabolic and signaling pathways, enzymes and metabolites mentioned in this review.

Redox systems have co-evolved with biological complexity, enabling synchronization across multi-scale networks [107]. Despite this, the redox interactome remains poorly studied due to methodological challenges and its high variability, which is context dependent. Certain redox molecules may play distinct roles in unrelated metabolic pathways [107], creating a landscape of functional plasticity. This complexity has led to discrepancies in interpreting the role of redox patterns in biological processes, particularly in cancer [108]. As a result, redox research is prone to a “Rashomon effect,” where seemingly contradictory conclusions arise from the same datasets and methods, potentially reinforcing untested preconceived ideas [108].

The study of cancer, as a prime example of a complex system, would benefit from greater consensus on classifying and understanding the causative roles of redox patterns in metabolic dynamics. Moreover, unexplored redox patterns could play critical roles in maintaining and stabilizing not only tumor states but also normal physiological conditions. Achieving clarity in this field may unlock new insights into cancer biology and therapeutic strategies. Studying cancer from a redox perspective could deepen our understanding of the broad range of metabolic effects that organisms may exhibit under non-cancerous conditions. Recent advances in imaging techniques have revealed unexpected metabolic heterogeneity within cancer, observed both in vivo and in vitro, across the same cancer types and even within different spatial regions of the same tumor [109]. This heterogeneity leads to metabolic flexibility, where cancer cells can choose from various metabolic substrates, and metabolic plasticity, which introduces alternative pathways for substrate metabolism [110]. Patients with similar mutational profiles often show vastly different outcomes due to the complexity of their underlying metabolic phenotypes [109]. Identifying whether these redox patterns, which appear critical in cancer dynamics, also exist under normal physiological conditions could provide insights similar to those obtained from genetic studies, where mutations once thought cancer-specific were later found in healthy tissues [111,112]. The difficulty in portraying a clear picture of metabolic features during cancer progression [113] is likely rooted in the open-ended evolutionary dynamics of metabolic activity in tumors as they continue to grow [114]. Consequently, redox patterns may serve as “selective filters” in cancer, contributing to the context-dependent flexibility observed in its metabolic rewiring (Figure 8). Surprisingly, this rewiring, once considered exclusive to pathological conditions like cancer, may also represent a normal aspect of physiological metabolism.

Supporting this view, a recent study suggested that metabolism in animal cells might be best understood as a bistable electric network, with redox imbalances playing a central role in the metabolic switches controlled by the master regulator of metabolism, mTOR [115]. Previous research also showed that redox systems in human cells can coexist in stable high-GSH/low-ROS and low-GSH/high-ROS states, exhibiting hysteresis, where cells tend to remain in these states before and after changes in ROS concentration [116]. However, further research is needed to understand how these low-scale patterns influence or contribute to higher-order features in organisms, such as cancer.

Despite their influence on higher-order processes, redox systems can also be influenced by these processes in a top-down causation regime. The organization of complex biological systems offers a paradigm for understanding how downward causation contributes to the evolution and stability of lower-scale structures and processes [117]. Paraphrasing Dobzhansky, it seems that nothing in complex systems makes sense except in the context of feedbacks, making the interaction between redox systems and cancer a prime example of how top-down and bottom-up causation might have evolved. Recent evidence has illuminated feedback mechanisms between redox systems and high-level regulatory elements in biological systems. For instance, the metabolic circadian clock has been shown to be sustained by interactions among redox elements such as thioredoxin, peroxiredoxin and NAD(P)H, independent of transcriptional activity [118,119], with NRF2 orchestrating the relationship between redox oscillations and circadian transcriptional activity [120]. Furthermore, the regulation of redox circadian patterns by glucose metabolism in human red blood cells highlights the role of feedback mechanisms in driving the contributions of redox systems to specific cellular dynamics [121].

Remarkably, redox patterns can, in turn, influence metabolic rewiring by directing glucose metabolism toward the pentose phosphate pathway, enhancing the reductive capacity of tumor cells. This metabolic shift supports anchorage-independent growth by targeting a specific cysteine residue on the glyceraldehyde-3-phosphate dehydrogenase (GAPDH) enzyme [122]. Additionally, redox oscillations have been implicated in cell cycle progression through changes in cytosolic pH [123], which may help explain their role in the intense metabolic demands of continuously growing tumors [114], positioning redox systems as central regulators of both metabolism and cellular dynamics. These findings are particularly relevant for understanding circadian clock regulation and its potential role in the development of pathological conditions.

Recent research has shown that circadian metabolism is highly coordinated across different tissues in organisms like mice, though it can be disrupted by sudden nutritional changes [124]. In cancer, this coordination is impaired, but it remains unclear whether disruptions in circadian rhythms are a cause or consequence of cancer dynamics [125]. The rhythmic patterns of certain redox molecules have been indirectly observed in patients [126], and these redox oscillations follow a distinctly different pattern in cancer compared to normal tissues [125]. Recently, circadian interventions modulated by redox systems have emerged as potential therapeutic strategies to reduce mortality in chronic diseases [127]. Alongside these developments, chronotherapy, which administers anti-tumor drugs at specific times, has gained attention in recent years as a way to improve cancer survival rates [128,129,130]. Thus, examining the regulation of circadian clocks by redox patterns and applying this knowledge to chronotherapy could significantly enhance our understanding of cancer metabolism and its regulation by redox systems.

The growing body of research on carcinogenesis highlights the critical role of feedback mechanisms between metabolism and redox systems. Consequently, an integrative approach to identifying significant patterns within this expanding dataset is crucial to building a comprehensive understanding of how cross-talk across different spatiotemporal scales directs biological dynamics to enhance flexibility and plasticity. Positioned at a foundational level in this evolutionary framework, redox systems may have paved the way for nested non-equilibrium processes that self-organize by harnessing the inherent noise characteristic of living systems. Future cancer research should recognize the need for a multi-scale approach, where interconnected dynamics replace isolated fields of study. The relationship between redox systems and metabolism is just one example that warrants deeper exploration. Investigating how redox systems regulate biological scales, in both cancer and other processes, promises to offer profound insights into the nature of life itself.

## Figures and Tables

**Figure 1 ijms-26-00498-f001:**
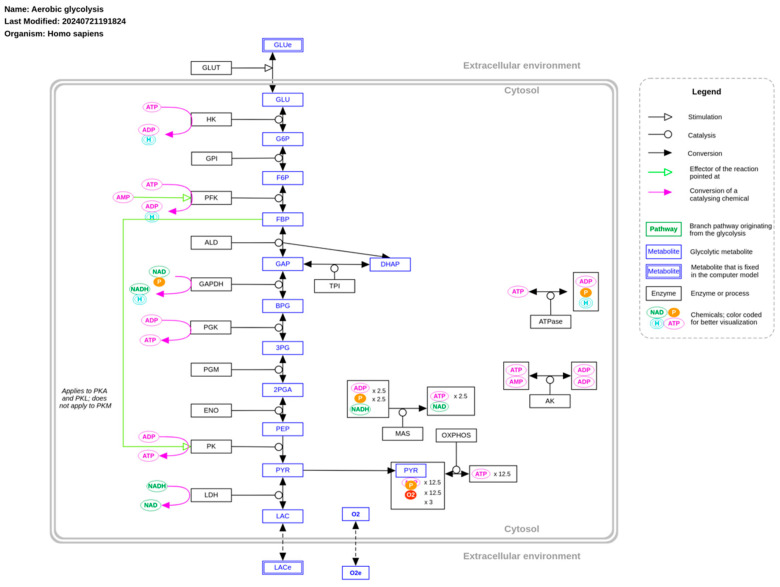
Aerobic glycolysis in cancer. Freely available and retrieved from: https://www.wikipathways.org/instance/WP4629 (accessed on 24 December 2024).

**Figure 2 ijms-26-00498-f002:**
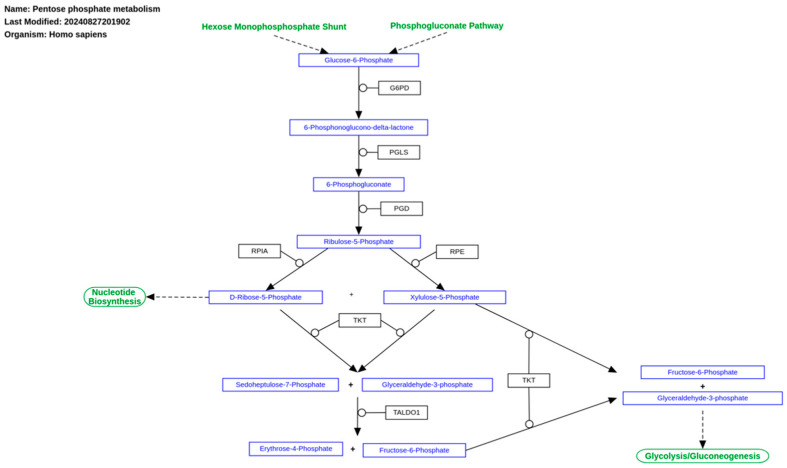
The pentose phosphate pathway and its connections with our metabolic pathways. Freely available and retrieved from: https://www.wikipathways.org/instance/WP134 (accessed on 24 December 2024).

**Figure 3 ijms-26-00498-f003:**
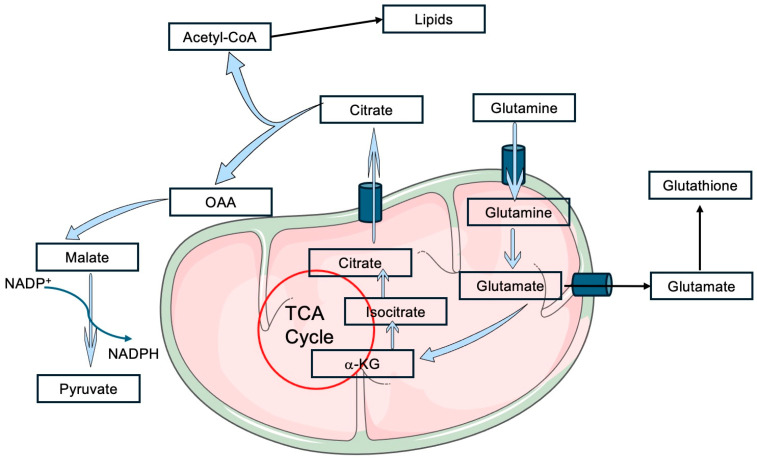
Glutamine catabolism via reductive carboxylation, malate dehydrogenase and malic enzyme I (blue arrows). Within the mitochondria, glutamine is substrate of glutaminase (either GLS or GLS2), releasing glutamate and ammonium. Glutamate as a substrate of glutamate dehydrogenase or aspartate aminotransferase yields α-ketoglutarate, an intermediate of the tricarboxylic acid (TCA) cycle. Glutamate can be used for glutathione biosynthesis. Under reductive conditions, reductive carboxylation occurs rendering citrate, that can be exported to the cytosol, where it is lysed by ATP-citrate lyase, releasing acetyl-CoA (for lipid biosynthesis) and oxaloacetate (OAA). In the cytosol, malate dehydrogenase transforms OAA into malate, which is a substrate of malic enzyme I, releasing pyruvate and generating reduced NADP. Figure drawn by the authors in a Power Point canvas and using some images freely available from SMART (Servier Medical Art).

**Figure 4 ijms-26-00498-f004:**
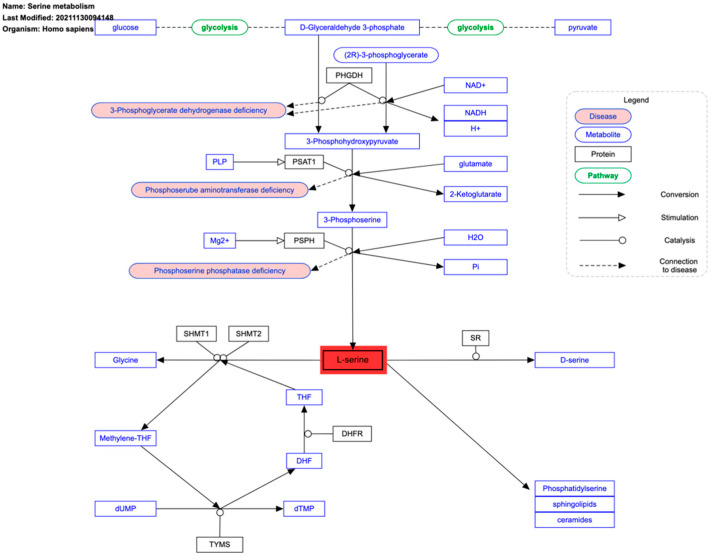
Serine metabolism pathway. Freely available and retrieved from: https://www.wikipathways.org/instance/WP4688 (accessed on 24 December 2024).

**Figure 5 ijms-26-00498-f005:**
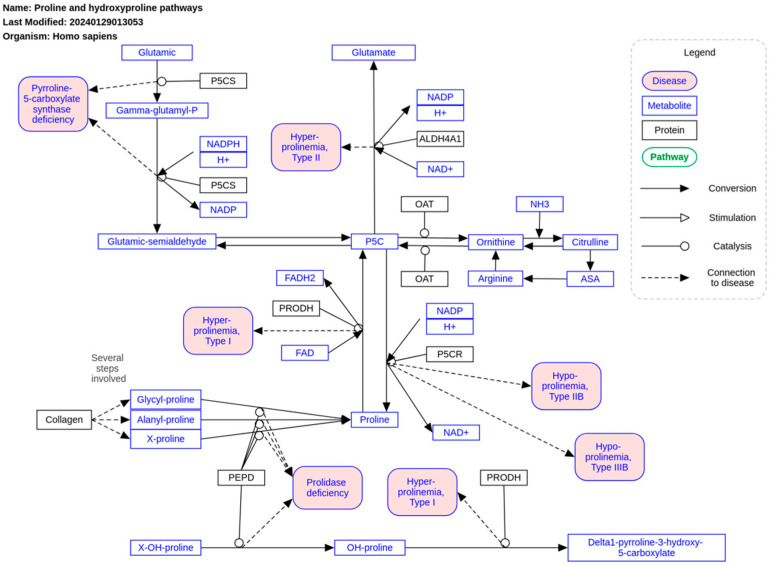
Proline metabolism pathway. Freely available and retrieved from: https://www.wikipathways.org/instance/WP5026 (accessed on 24 December 2024).

**Figure 6 ijms-26-00498-f006:**
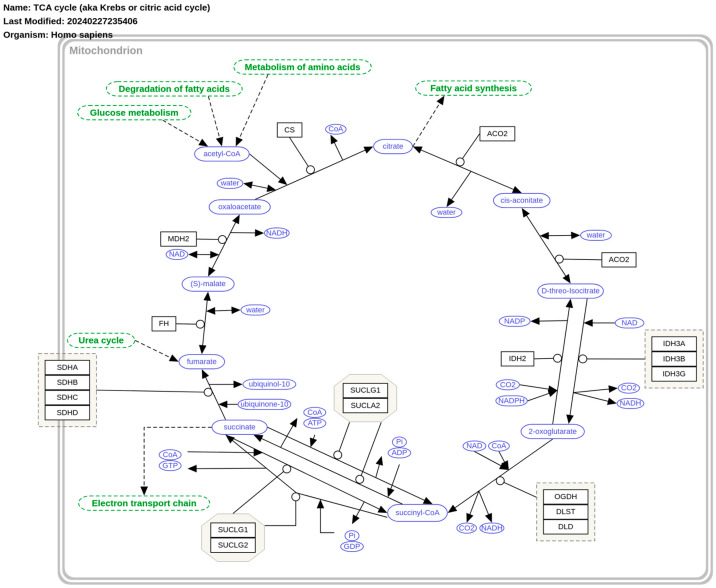
The tricarboxylic acid (TCA) cycle or Krebs cycle. Freely available and retrieved from: https://www.wikipathways.org/instance/WP78 (accessed on 24 December 2024).

**Figure 7 ijms-26-00498-f007:**
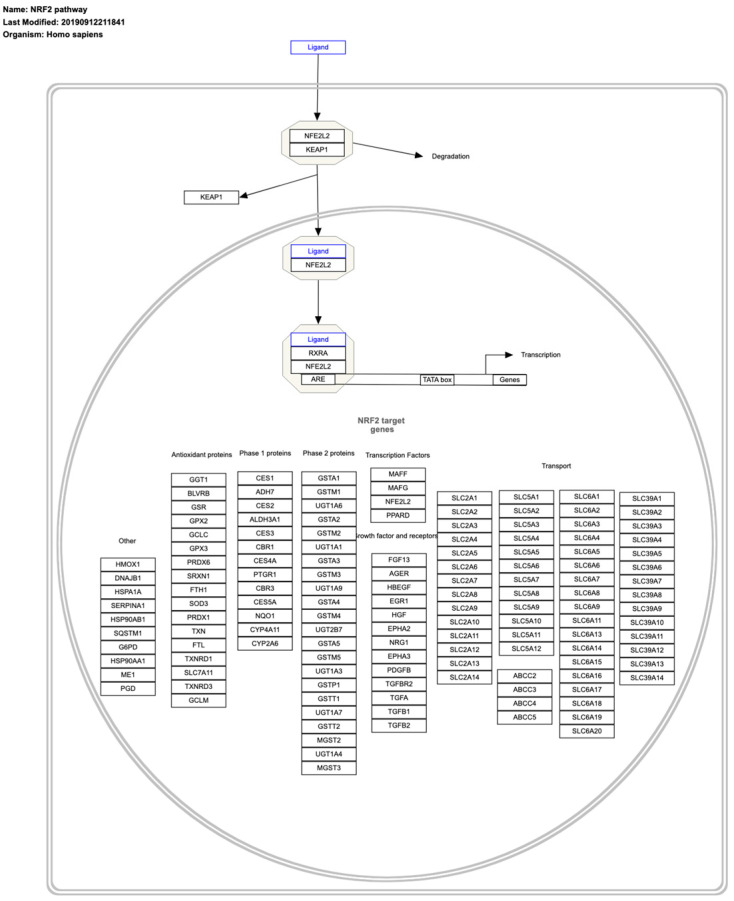
NRF2 transcription factor signaling pathway. Freely available and retrieved from: https://www.wikipathways.org/instance/WP2884 (accessed on 24 December 2024).

**Figure 8 ijms-26-00498-f008:**
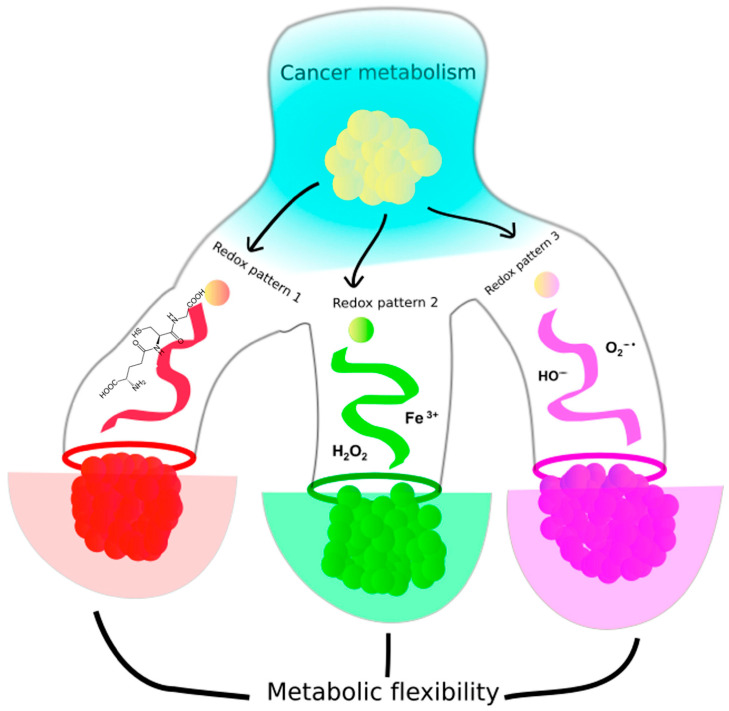
Cancer metabolism of an originally homogeneous phenotype might be driven towards different metabolic states influenced by the variety of redox patterns found in the cancer environment. The flexibility exhibited by cancer metabolism follows different pathways on the metabolic landscape if different combinations of redox pairs are more abundant than others.

**Table 1 ijms-26-00498-t001:** Main metabolic and signaling pathways, enzymes and metabolites mentioned in this review.

Metabolic/Signaling Pathway	Enzyme ^1^/Metabolite	Comments
Aerobic glycolysis	PKM2	Enhanced in cancer
	Lactate	Increased production
Pentose phosphate pathway	G6PD TLK	Regulated by cellular ROS through Nrf2 activation

Glutamine catabolism	Glutamine	TCA ^2^ anaplerosis
	Malic enzyme I	Reductive TCA
mTOR		Translocation to mitochondria

Serine metabolism	PHGDH	Overexpressed in hypoxia
Proline/pyrroline-5-carboxylate cycle		In lung cancer, it supports glycolysis
TCA	IDH SDH FH	Its deficiency is pro-oxidative Its deficiency is prooxidative A more reductive cell state Cell
Heme catabolism	HO-1 + BVR	It prevents ROS formation through the Fenton reaction
Nrf2		A central regulator of cell redox

^1^ PKM2: pyruvate kinase M2; G6PD: glucose-6-phosphate dehydrogenase; TLK: transketolase; PHGDH: 3-phosphoglycerate dehydrogenase; IDH: isocitrate dehydrogenase; SDN: succinate dehydrogenase; FH: fumarase. ^2^ TCA: tricarboxylic acid cycle (Krebs’ cycle).

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
