# Peer review of "Metabolic Reprogramming at the Edge of Redox: Connections Between Metabolic Reprogramming and Cancer Redox State"

_ijms, 2025, doi:10.3390/ijms26020498_

Round 1
Reviewer 1 Report
Comments and Suggestions for Authors
This review on redox regulation of metabolic pathways has useful information and would be of interest to many readers. However it would benefit from editing by a native English speaker. Long sentence structures could be divided into smaller sentences and improve accessibility and style. Very dense as written.
Page 1: ‘The metabolic capacity of tumors is well documented in the case of glucose, fatty acids or glutamine flows [10], but the true role of redox systems in the process of tumor metabolic reprogramming has yet to be determined.’ Many aspects of redox state regulation of metabolism have been defined over the last two decades. Role(s) can be plural and it could be said that redox sensitive mechanisms could be better characterized at a molecular level.
Page 2: ‘It remains to be seen whether in tumor metabolism (and even in the metabolism of untransformed cells), it is possible to find phenotypes similar to those observed in yeasts, and which could be key to understanding metabolic versatility, as well as phenotypic heterogeneity, which is typical of tumors.’ Rephrase this please. It’s another run-on sentence where the meaning is obscured.
Page 2: Again, rephrase this: ‘This metabolic change has been reversed in docetaxel-resistant PCa cells, moving from a metabolism based on oxidative phosphorylation to one where weight is carried by glycolytic flow, through the re-expression of a miRNA, miR-205, whose inhibition is associated with an increased respiratory phenotype, and which makes these cells sensitive to docetaxel again [20].’
Page 2: ‘It should be stressed that a great heterogeneity has been observed in the management of oxidative stress by different cells from a panel of breast cancer cell lines [22], and that a similar heterogeneity should be expected in other cancer cell types. It is also noteworthy that metabolic reprogramming with an increased dependence on glycolysis aids to main tain redox homeostasis during metastasis in breast cancer cells [23].’ It should be pointed out that metabolic regulation of cell lines can be vastly different from the same tissue types in vivo.
Figure 1 is very simplistic and adds little as is. Figures defining the metabolic pathways in each manuscript section would be more informative and help the flow of the review.
Page 3: This is very unclear. ‘It remains to be seen whether there is a relationship between Akt expression levels and the translocation of mTOR to the mitochondria, modifying glutamine levels, since in both cases there is an increase in OXPHOS accompanied by mechanisms aimed at maintaining the redox homeostasis of tumor cells, which is further evidence that tumor cells modify their redox state to deal with external or internal disturbances, respectively.’ Akt expression and activation state are separable and only Akt activation drives TORC1 activity. Referring to Akt phosphorylation state (PDK1 and TORC2 regulation) would be more informative.
Page 6: The hyperlink reference needs to be formatted.

Needs substantial editing/rewriting.
Author Response
REPLY TO REVIEWER 1 (Reviewer’s comments in black. Our reply in blue)
This review on redox regulation of metabolic pathways has useful information and would be of interest to many readers. However it would benefit from editing by a native English speaker. Long sentence structures could be divided into smaller sentences and improve accessibility and style. Very dense as written.
First of all, thank you for your kind comments, suggestions, and criticism. Regarding the writing style, we have rewritten fully the manuscript improving its accessibility and style, with smaller sentences and clearer structure, without changing the final meaning and messages.
Page 1: ‘The metabolic capacity of tumors is well documented in the case of glucose, fatty acids or glutamine flows [10], but the true role of redox systems in the process of tumor metabolic reprogramming has yet to be determined. ’ Many aspects of redox state regulation of metabolism have been defined over the last two decades. Role(s) can be plural and it could be said that redox sensitive mechanisms could be better characterized at a molecular level.
Done.
Page 2: ‘It remains to be seen whether in tumor metabolism (and even in the metabolism of untransformed cells), it is possible to find phenotypes similar to those observed in yeasts, and which could be key to understanding metabolic versatility, as well as phenotypic heterogeneity, which is typical of tumors. ’ Rephrase this please. It’s another run-on sentence where the meaning is obscured.
Changed. Rephrased.
Page 2: Again, rephrase this: ‘This metabolic change has been reversed in docetaxel-resistant PCa cells, moving from a metabolism based on oxidative phosphorylation to one where weight is carried by glycolytic flow, through the re-expression of a miRNA, miR-205, whose inhibition is associated with an increased respiratory phenotype, and which makes these cells sensitive to docetaxel again [20]. ’
Changed. Rephrased.
Page 2: ‘It should be stressed that a great heterogeneity has been observed in the management of oxidative stress by different cells from a panel of breast cancer cell lines [22], and that a similar heterogeneity should be expected in other cancer cell types. It is also noteworthy that metabolic reprogramming with an increased dependence on glycolysis aids to main tain redox homeostasis during metastasis in breast cancer cells
[23]. ’ It should be pointed out that metabolic regulation of cell lines can be vastly different
from the same tissue types in vivo.
Changed. Rephrased.
Figure 1 is very simplistic and adds little as is. Figures defining the metabolic pathways in each manuscript section would be more informative and help the flow of the review.
Done. Figure 1 has been eliminated and we have introduced new figures summarizing the main metabolic and signaling pathways commented in the manuscript.
Page 3: This is very unclear. ‘It remains to be seen whether there is a relationship between Akt expression levels and the translocation of mTOR to the mitochondria, modifying glutamine levels, since in both cases there is an increase in OXPHOS accompanied by mechanisms aimed at maintaining the redox homeostasis of tumor cells, which is further evidence that tumor cells modify their redox state to deal with external or internal disturbances, respectively. ’ Akt expression and activation state are separable and only Akt activation drives TORC1 activity. Referring to Akt phosphorylation state (PDK1 and TORC2 regulation) would be more informative.
Changed. Rephrased.
Page 6: The hyperlink reference needs to be formatted.
Done. Now the link works properly.
Reviewer 2 Report
Comments and Suggestions for Authors
The manuscript presents the interplay between metabolism and redox systems in carcinogenesis. It is a very generous and up-to-date topic that can open the way to new approaches regarding the development of new antitumor therapy. The manuscript presents interest to the readers of this journal and can be published after a minor revision.
as comments/suggestions:
1. Assigning a title closer to the content of the article (redox reactions from the body and their involvement in the carcinogenesis process are presented)
2. The introduction of schemes or images representative of each category of redox systems would help less experienced readers to better understand the content of this article
3. The manuscript could be completed with the already existing discoveries regarding the categories of substances that act on these redox systems at the level of the body involved in the initiation of the tumor process (e.g. metal complexes - Fenton or Haber Weiss type reactions)
Author Response
REPLY TO REVIEWER 2 (Reviewer’s comments in black. Our reply in blue)
The manuscript presents the interplay between metabolism and redox systems in carcinogenesis. It is a very generous and up-to-date topic that can open the way to new approaches regarding the development of new antitumor therapy. The manuscript presents interest to the readers of this journal and can be published after a minor revision.
First of all, thank you for your kind comments, suggestions, and positive criticism. Regarding the writing style, in response to a suggestion raised by reviewer 1 we have rewritten fully the manuscript improving its accessibility and style, with smaller sentences and clearer structure, without changing the final meaning and messages.
As comments/suggestions:
- Assigning a title closer to the content of the article (redox reactions from the body and their involvement in the carcinogenesis process are presented)
Done. According to your suggestion, we have changed the title to a new one closer to the content of the article: Metabolic Reprogramming at the Edge of Redox: Connections Between Metabolic Reprogramming and Cancer Redox State.
- The introduction of schemes or images representative of each category of redox systems would help less experienced readers to better understand the content of this article
Done. According to your own suggestion and to another one raised by reviewer 1, we have introduced a set of figures with schemes of the main metabolic and signaling pathways mentioned in the manuscript.
- The manuscript could be completed with the already existing discoveries regarding the categories of substances that act on these redox systems at the level of the body involved in the initiation of the tumor process (e.g. metal complexes - Fenton or Haber Weiss type reactions)
Thank you for your suggestion, but we prefer to maintain the focus in the connections between metabolic reprogramming and cancer redox state.
Reviewer 3 Report
Comments and Suggestions for Authors
The manuscript entitled “Metabolic Reprogramming at the Edge of Redox” by Jose S et al., was well written and easy to follow. I have the following suggestions/comments
1. It would be helpful if a schematic can be added to each section connecting the linkage to cancers and the pathways related to that specific metabolite. This gives better perspective of the summary.
2. Most of these metabolites are very well studied and understood, what is the new information that you are trying to add to the current manuscript, as it is not so clear?
3. Please add new information on various trials related to metabolites and connect them with redox states. Please try to surround the perspective based on the patient data as that can bring more enthusiasm in audience and helps strengthen the overall understanding
4. It is well known of these metabolites connecting to redox biology, please try and connecting them in cellular/disease context and provide your perspectives on targeting them.
5. Adding information on various cancers and their available metabolic signatures from previous studies can provide a broader scope of the literature.
6. Listing various cancers and their expressions connecting to disease outcome could be helpful.
7. The recent advancement in the metabolic signatures of exosomes can bring in a new dimension to our understanding on redox biology, adding such information may be helpful too.
8. There are too many grammatical and spelling errors that need to be addressed please modify them accordingly
Author Response
REPLY TO REVIEWER 3 (Reviewer’s comments in black. Our reply in blue)
The manuscript entitled “Metabolic Reprogramming at the Edge of Redox” by Jose S et al., was well written and easy to follow. I have the following suggestions/comments
First of all, thank you for your kind comments, suggestions, and positive criticism. Regarding the writing style, in response to a suggestion raised by reviewer 1 we have rewritten fully the manuscript improving its accessibility and style, with smaller sentences and clearer structure, without changing the final meaning and messages.
- It would be helpful if a schematic can be added to each section connecting the linkage to cancers and the pathways related to that specific metabolite. This gives better perspective of the summary.
Done. According to your own suggestion and those raised by reviewers 1 and 2, we have introduced a set of figures with schemes of the main metabolic and signaling pathways mentioned in the manuscript.
- Most of these metabolites are very well studied and understood, what is the new information that you are trying to add to the current manuscript, as it is not so clear?
Since this is a review manuscript, of course we do not aim to add any new information, but to provide a comprehensive view of our current knowledge of the connections between metabolic reprogramming and cancer redox state. To make our message clearer, in response to a suggestion raised by reviewer 1 we have rewritten fully the manuscript improving its accessibility and style, with smaller sentences and clearer structure, without changing the final meaning and messages.
- Please add new information on various trials related to metabolites and connect them with redox states. Please try to surround the perspective based on the patient data as that can bring more enthusiasm in audience and helps strengthen the overall understanding
- It is well known of these metabolites connecting to redox biology, please try and connecting them in cellular/disease context and provide your perspectives on targeting them.
- Adding information on various cancers and their available metabolic signatures from previous studies can provide a broader scope of the literature.
- Listing various cancers and their expressions connecting to disease outcome could be helpful.
- The recent advancement in the metabolic signatures of exosomes can bring in a new dimension to our understanding on redox biology, adding such information may be helpful too.
Points 3-8. We appreciate your suggestions. However, all the changes required to introduces your suggestion would render a much longer and a very different review, very far from our aim to review the current knowledge on the connections between metabolic reprogramming and cancer redox state.
- There are too many grammatical and spelling errors that need to be addressed please modify them accordingly.
Done. We have rewritten fully the manuscript correcting grammatical and spelling errors, improving its accessibility and style, with smaller sentences and clearer structure, without changing the final meaning and messages.
Reviewer 4 Report
Comments and Suggestions for Authors
The manuscript "Metabolic Reprogramming at the Edge of Redox" by Serrano et al summarized connections between metabolism and cancer redox state. Personally, I think that this is very brief review article and is not yet ready for publication. This manuscript is needed to have more references, figures, tables for actual review, for example, signalling pathways and the effects. The cited papers should be updated including published review articles. The URL, website should be removed or as reference. The resolution of the figure is too low. The references should be formatted in mdpi style.
Author Response
REPLY TO REVIEWER 4 (Reviewer’s comments in black. Our reply in blue)
The manuscript "Metabolic Reprogramming at the Edge of Redox" by Serrano et al summarized connections between metabolism and cancer redox state. Personally, I think that this is very brief review article and is not yet ready for publication. This manuscript is needed to have more references, figures, tables for actual review, for example, signalling pathways and the effects. The cited papers should be updated including published review articles. The URL, website should be removed or as reference. The resolution of the figure is too low. The references should be formatted in mdpi style.
Thank you for your comments. We have reviewed the contents of IJMS in the last month and we have found reviews that are longer but also other reviews that are shorter than our own manuscript. The same applies for the number of references: IJMS publishes reviews with less and with more references that the 130 references we cite in our manuscript. According to suggestions raised by reviewers 1-3, we have introduced a set of figures with schemes of the main metabolic and signaling pathways mentioned in the manuscript. Figure 1 has been eliminated. Figure 2 in the original manuscript has proper resolution. The same applies for the new figures. The URL included in the maintext is an announcement, not a reference; for this reason, we keep it (modified, since in the original manuscript it was wrongly written) in the maintext. We have revised all the references and corrected those that were not formatted in mdpi style.
Round 2
Reviewer 3 Report
Comments and Suggestions for Authors
The authors tried to address several comments but the sections that add additional information besides the comprehensive information available in literature is missing.
It would be encouraging to see a table to metabolites and signatures that are available in redox world to provide a bigger picture of the understanding in field.
Author Response
REPLY TO REVIEWER 3 (Reviewer’s comments in black. Our reply in blue)
The authors tried to address several comments but the sections that add additional information besides the comprehensive information available in literature is missing.
It would be encouraging to see a table to metabolites and signatures that are available in redox world to provide a bigger picture of the understanding in field.
First of all, thank you for your kind comments, suggestions, and positive criticism.
Regarding the first paragraph, we do not fully understand you. In any case, our review focuses on the connections between metabolic reprogramming and cancer redox state. We are aware that much more information is available on cancer redox state, but this is not the aim of our review.
Regarding your suggestion, now we have introduced a Table 1 summarizing the main metabolic and signaling pathways, enzymes and metabolites mentioned in this review.
Reviewer 4 Report
Comments and Suggestions for Authors
Accept in present form
Author Response
REPLY TO REVIEWER 4 (Reviewer’s comments in black. Our reply in blue)
Accept in present form
Thank you for your positive evaluation of our manuscript.